# Omega-3 Fatty Acid Intake during Pregnancy and Child Neuropsychological Development: A Multi-Centre Population-Based Birth Cohort Study in Spain

**DOI:** 10.3390/nu14030518

**Published:** 2022-01-25

**Authors:** Hana Tahaei, Florence Gignac, Ariadna Pinar, Silvia Fernandez-Barrés, Dora Romaguera, Jesus Vioque, Loreto Santa-Marina, Mikel Subiza-Pérez, Sabrina Llop, Raquel Soler-Blasco, Victoria Arija, Jordi Salas-Salvadó, Adonina Tardón, Isolina Riaño-Galán, Jordi Sunyer, Monica Guxens, Jordi Julvez

**Affiliations:** 1ISGlobal, 08003 Barcelona, Spain; hana.tahaei44@gmail.com (H.T.); florence.gignac@isglobal.org (F.G.); ari.pinar01@estudiant.upf.edu (A.P.); silvia.fernandez@isglobal.org (S.F.-B.); dora.romaguera@isglobal.org (D.R.); jordi.sunyer@isglobal.org (J.S.); monica.guxens@isglobal.org (M.G.); 2Department of Experimental and Health Sciences, Universitat Pompeu Fabra (UPF), 08003 Barcelona, Spain; 3Institut d’Investigació Sanitària Pere Virgili (IISPV), Hospital Universitari Sant Joan de Reus, 43204 Reus, Spain; jordi.salas@urv.cat; 4Health Research Institute of the Balearic Islands (IdISBa), 07120 Palma de Mallorca, Spain; 5Centro de Investigación Biomédica en Red Fisiopatología de la Obesidad y la Nutrición (CIBEROBN), Institute of Health Carlos III, 28029 Madrid, Spain; 6Instituto de Investigación Sanitaria y Biomédica de Alicante, Universidad Miguel Hernández (ISABIAL-UMH), 46020 Alicante, Spain; vioque@umh.es; 7Centro de Investigacion Biomedica en Red Epidemiologia y Salud Pública (CIBERESP), Instituto de Salud Carlos III (ISCIII), 28029 Madrid, Spain; ambien4ss-san@euskadi.eus (L.S.-M.); mikel.subiza@ehu.eus (M.S.-P.); atardon@uniovi.es (A.T.); isolinariano@gmail.com (I.R.-G.); 8Ministry of Health of the Basque Government, Sub-Directorate for Public Health and Addictions of Gipuzkoa, 20013 San Sebastian, Spain; 9Group of Environmental Epidemiology and Child Development, Biodonostia Health Research Institute, 20014 San Sebastian, Spain; 10Epidemiology and Environmental Health Joint Research Unit, FISABIO—Universitat Jaume I—Universitat de València, 46020 Valencia, Spain; llop_sab@gva.es (S.L.); raquel.Soler@uv.es (R.S.-B.); 11Universitat Rovira i Virgili (URV), Nutrition and Mental Health Research Group (NUTRISAM), 43201 Reus, Spain; victoria.arija@urv.cat; 12Universitat Rovira i Virgili (URV), Departament de Bioquímica i Biotecnologia, Unitat de Nutrició Humana, 43201 Reus, Spain; 13Departamento de Medicina, Universidad de Oviedo, 33006 Oviedo, Spain; 14IUOPA-ISPA, Avda. Hospital Universitario s/n, 33011 Oviedo, Spain; 15Servicio de Pediatría, Endocrinología Pediátrica, HUCA, Roma Avenue S/n, 33001 Oviedo, Spain; 16Department of Child and Adolescent Psychiatry/Psychology, Erasmus MC, University Medical Centre, 3015 CN Rotterdam, The Netherlands

**Keywords:** maternal diet, children, neurodevelopment, omega-3 fatty acids, population-based cohort

## Abstract

Background: There are few studies that look at the intake of all types of omega-3 polyunsaturated fatty acids (n-3 PUFAs) during the different stages of pregnancy along with a long-term neuropsychological follow-up of the child. This study aims to explore the association between maternal n-3 PUFA intake during two periods of pregnancy and the child’s neuropsychological scores at different ages. Methods: Prospective data were obtained for 2644 pregnant women recruited between 2004 and 2008 in population-based birth cohorts in Spain. Maternal n-3 PUFA intake during the first and third trimester of pregnancy was estimated using validated food frequency questionnaires. Child neuropsychological functions were assessed using Bayley Scales of Infant Development version one (BSID) at 1 year old, the McCarthy Scale of Children’s Abilities (MSCA) at 4 years old, and the Attention Network Test (ANT) at 7 years old. Data were analysed using multivariate linear regression models and adjusted for potential covariates, such as maternal social class, education, cohort location, alcohol consumption, smoking, breastfeeding duration, and energy intake. Results: Compared to participants in the lowest quartile (<1.262 g/day) of n-3 PUFA consumption during the first trimester, those in the highest quartile (>1.657 g/day) had a 2.26 points (95% confidence interval (CI): 0.41, 4.11) higher MSCA general cognitive score, a 2.48 points (95% CI: 0.53, 4.43) higher MSCA verbal score, and a 2.06 points (95% CI: 0.166, 3.95) higher MSCA executive function score, and a 11.52 milliseconds (95% CI: −22.95, −0.09) lower ANT hit reaction time standard error. In the third pregnancy trimester, the associations were weaker. Conclusions: Positive associations between n-3 PUFA intake during early pregnancy and child neuropsychological functions at 4 and 7 years of age were found, and further clinical research is needed to confirm these findings.

## 1. Introduction

One of the most important objectives in public health is to improve nutrition during pregnancy, and this has been addressed in the United Nations 2030 Sustainable Development Goals and the 2025 Global Nutrition Targets, which recognize that better nutrition is essential in the steps towards better global health [1,2]. Women adhering to dietary recommendations during pregnancy may lead to an optimal growth and health outcome of the infant [3]. One recommendation that has been of growing interest is the provision of an adequate intake of omega-3 polyunsaturated fatty acids (n-3 PUFAs). This type of PUFA is found in three forms: docosahexaenoic acid (DHA), alpha-linolenic acid (ALA), and eicosapentaenoic acid (EPA). ALA is an essential fatty acid, as the body cannot synthetize it due to the lack of required enzymes, and it must be obtained through food or supplements. Food sources of ALA include nuts, seeds, and vegetable oils. It can be converted in the body into the form of EPA and DHA. However, this pathway is very limited (1%) and can vary among individuals, particularly during pregnancy. Therefore, it is recommended that DHA and EPA are obtained directly from food sources, such as fatty oily fish, including salmon, mackerel, and sardines, and/or supplementation [4,5,6].

There is currently no clear set recommended dietary allowance (RDA) for n-3 PUFAs across all lifespans, including pregnancy. Daily adequate intake (AI) of ALA during pregnancy is set at 1.4 g [5]. The European Food Safety Authority recommends a daily AI of 250 mg of EPA and DHA, or the consumption of two servings of fish per week for the general population and an extra 100–200 mg daily of DHA during pregnancy [4]. The greater need during pregnancy is due to findings that increased amounts of n-3 PUFAs are required by the foetus [6,7]. This adequate intake of n-3 PUFAs has been shown to be vital for the formation of cell membranes in certain tissues, including the brain tissue, and the development of neurons and their synaptic activity [8,9].

Optimal foetal brain and neuron development are related to the adequate maternal intake of these nutrients during pregnancy [10,11,12]. There is growing evidence that the proportion of n-3 PUFAs in maternal plasma is positively correlated with the proportion in foetal plasma [7]. Furthermore, it has been shown that the developing brain is more susceptible to n-3 PUFA deficiency than the mature brain [5,8]. The developing brain, particularly during pregnancy and the first year of life, requires the appropriate amount of essential nutrients for structural growth and the formation of synaptic connections. The inhibition of these developing processes may lead to long-term functional consequences for the brain [5,13]. Furthermore, some research has shown that n-3 PUFA composition in human milk is dependent on not just short-term maternal diet, such as a few weeks, but also long-term intake, such as a few months, prior to the postpartum period [13].

Nowadays, there is some research about the maternal dietary intake of n-3 PUFAs (including fish and nuts) and its impact on child neurodevelopment [9,13,14,15,16]. However, longitudinal cohort studies with large sample size are limited. To the best of our knowledge, there is a strong need to conduct extensive studies with repeated estimations of food consumption during pregnancy, specifically looking at the maternal dietary intake of all types of n-3 PUFAs during the first and third trimesters of pregnancy, along with neuropsychological outcomes via various neuropsychological tests at different ages of their child.

In the present study, we assessed associations between the maternal dietary intake of n-3 PUFAs during the first and third trimester of pregnancy, and child neuropsychological outcomes at 1, 4, and 7 years of age.

## 2. Subjects and Methods

### 2.1. Study Population

This study was based on a birth cohort that included participants from different regions of Spain under the INMA (Infancia y Media Ambiente (Environment and Childhood)) project. Each study site (region) followed the same study design, questionnaires, and protocol assessments. The participants in this study were from Valencia, Sabadell, Asturias, and Gipuskoa, with recruitment occurring from 2004 to 2008. Eligible women who presented for antenatal care in main public hospitals or health centres were first given written information regarding the INMA project, followed by their consent if they accepted to participate. Their children were enrolled upon birth and administrated neuropsychological scales and computer-based tests up to the 7th-year visit. The number of participants included in the study is outlined in Figure 1. The criteria of inclusion of the mothers ensured that they were living in the study area (specific to each county), were at least 16 years of age, had no communication problems, had intention to give birth at the reference hospital, were carrying a singleton pregnancy, and were not partaking in any program for assisted reproduction [17]. Of those eligible, just over half agreed to be recruited, and the common reasons for not partaking in the study included the report of no interest or no time [17]. Participation flowchart is presented in Figure 1. A total of 2644 pregnant women were recruited. Data regarding n-3 PUFA intake during the first trimester and third trimester were obtained from 97.8% and 92.4%, respectively, of the originally recruited women. Throughout the pregnancy, common reasons for being excluded were miscarriages, withdrawals, or foetal deaths. Some did not participate during pregnancy, but were assessed again when delivering their child. However, for this current study they were excluded due to missing data from the food frequency questionnaire (FFQ) that was administered during pregnancy. The study included 2318 children at around 1 year of age, 2030 children at around 4 years of age, and 1737 at around 7 years of age. However, after excluding subjects with missing data or incomplete neuropsychological testing at each follow-up, the numbers were reduced to 2104, 1713, and 1574, respectively.

### 2.2. Exposure Information

A 101-item semi-quantitative food frequency questionnaire (FFQ) was administered by trained interviewers, and this was used to assess diet twice during pregnancy, in the first trimester (weeks 10–13) and third trimester (weeks 28–32). The FFQ was an adapted version of Willett’s questionnaire that we developed and validated among pregnant women from the Valencia cohort [18]. Dietary intakes referred to the period from conception to the first assessment, and from the first to the second assessment, respectively. Standard units or serving sizes were specified for each food item. We calculated nutrient intakes by multiplying the frequency of use for each food item by the nutrient composition of the portion size specified on the FFQ, and then by the addition across all foods to obtain a total nutrient intake for each individual. We used the residual method to estimate calorie-adjusted values for nutrient intakes [19]. Nutrient estimates from 101 food items, in grams per day, were obtained, and this data were used to assess usual maternal dietary n-3 PUFA intake. These were used in nine different frequency categories, ranging from ‘never/less than once a month’ to ‘more than 6 times per day’, both in servings per day and grams per day. The intake of n-3 PUFAs was obtained by adding the intake frequencies of the fatty acids at 22:5 (EPA), 22:6 (DHA) and 18:3 (ALA). The main source for EPA and DHA was seafood products identified through eight items of the food frequency questionnaire (non-canned fatty fish; lean fish; smoked/salted fish; molluscs; shrimp, prawn, and crayfish; octopus, baby squid, and squid; fatty fish canned in oil; fatty fish canned in salted water). Major sources of ALA were identified through four food items (nuts; margarine; corn oil and sunflower oil; olive oil).

The overall reproducibility and validity of the FFQ was acceptable for the intake of most nutrients and food groups. The reproducibility of the FFQ for the intake of PUFA (r = 0.50) and the n-3 PUFAs (r = 0.53) was acceptably good [18]. The biochemical validity of the FFQ was also assessed in the pregnant women of the INMA Sabadell cohort study [20]; statistically significant correlations (*p* < 0.01) were observed between plasma concentrations and dietary intakes for n-3 PUFA (r = 0.19), EPA (r = 0.33), and DHA (r = 0.28). Lower correlations are usually found in the literature when comparing nutrient intakes with plasma concentrations [21]. Thus, the overall validity of the FFQ to assess n-3 PUFA intake may be considered to be acceptable.

### 2.3. Co-Variable Information

Most of the covariate data that we considered, based on their impact on cognitive endpoints, and before selecting a reduced group of covariates for the multivariate analyses, were measured using standardized questionnaires that were conducted (in person) during the first and third pregnancy trimesters. We gathered information on sociodemographic characteristics and other lifestyle factors, such as parental education, social class, smoking, and alcohol consumption. Maternal dietary intake for other nutrients that may be associated with neuropsychological outcomes of children [11] were also explored, which includes protein, iron, folate, vitamin B12, and supplements of omega-3 fatty acids. To further evaluate participant’s diet quality in relevance to the Mediterranean diet, the alternative Mediterranean diet (aMED) score was derived from the FFQs [16]. PUFA concentrations (% among total fatty acids), collected from a subsample of participants, were then analysed in cord plasma (*n* = 947) via fast-gas chromatography. Individual cord blood ALA, EPA, DHA, linoleic acid (LA), and arachidonic acid (AA) were measured and expressed as a percentage of the total fatty acids. Cord blood total mercury (µg/L) concentration was analysed via thermal decomposition, amalgamation, and atomic absorption spectrometry by using a single purpose AMA-254 advanced mercury analyser (LECO Corporation, St. Joseph, MI, USA). Postnatal questionnaires were also completed to obtain information regarding lifestyle habits, including breastfeeding and smoking in the presence of the child. Child sex and weight at birth were obtained from clinical records. Postnatal psychometric information regarding the mother’s mental health and mother’s proxy of verbal intelligence quotient (IQ) were obtained at the 4th-year visit.

### 2.4. Neuropsychological Assessments

A common protocol was followed for psychological evaluation and scoring, and each centre/region used the same tests and versions. The psychologist of each centre/region completed mandatory training, and quality control evaluations were applied at each follow-up visit. The psychological examiners were blind to the exposure of interest. For this study, we selected the main cognitive outcomes assessed. Child neuropsychological outcomes were assessed using scores from the Bayley Scales of Infant Development version one (BSID) at 1 year of age [15], the McCarthy Scale of Children’s Abilities (MSCA) at 4 years of age [15,16], and the computer-based Attention Network Test (ANT) at 7 years of age [16].

The BSID consists of a series of subtests for assessing mental and psychomotor development for children from 1 to 30 months of age. The main mental and motor development scores were reviewed, where tasks were administered by several trained neuropsychologists to ensure internal validity and reliability [15,16]. These raw scores were standardized by age at assessment [22] and centred to a mean of 100, with a standard deviation of 15. Mental development scores assessed outcomes such as language, memory, and visual performance, while motor performances include assessing tasks such as sitting and climbing stairs.

The MSCA was used for the evaluation of psychomotor and cognitive performance. To accommodate for the Spanish population, the INMA project obtained a standardized version of this test from an organization called TEA that specializes in psychometric tests for Spain [23]. For this study, standardized MSCA scores with internal consistency and reliability from seven outcomes were assessed: general cognitive index, verbal, perceptive performance, numerical, memory, motor, and executive function. These scores were also centred to a mean of 100, with a standard deviation of 15. Detailed information about the MSCA can be found elsewhere [15,16,23].

The ANT is a computer-based test that measures attention function based on alerting, orienting, and conflict paradigms. The test evaluates hit reaction times based on different stimuli and clues. Hit reaction time standard error was considered to be a main measurement of sustained attention variability [16]. The ANT test is described in further detail elsewhere [24].

### 2.5. Statistical Analysis

The n-3 PUFAs and all the other nutrients were adjusted for energy intake using the residual method [25]. Per nutritional epidemiology reporting guidelines [26], the continuous n-3 PUFA intake from the first and third trimester that we obtained were categorized into quartiles. The values for the equally divided quartiles for the first trimester were as follows: 1st quartile: <1.262 g/day, 2nd quartile: 1.262 g–1.440 g/day, 3rd quartile: 1.440 g–1.657 g/day, and 4th quartile: >1.657 g/day. The values for the quartiles from the third trimester were as follows: 1st quartile: <1.219 g, 2nd quartile: 1.219 g–1.399 g, 3rd quartile: 1.399 g–1.599 g, and 4th quartile: >1.599 g. Mothers reported slightly higher levels of n-3 PUFA intake during the first pregnancy trimester questionnaire than in the third trimester questionnaire, but they were highly correlated (rho = 0.43). A descriptive analysis was developed to present each quartile and see if there were any significant differences between the covariate variables, including maternal, paternal, and child characteristics, chosen from a literature review [11,27,28]. An ANOVA was used to compare means of continuous variables against the categorical n-3 PUFA intake values; otherwise, a Chi-square test was used for categorical variables.

Associations between the daily maternal intake of n-3 PUFAs (in quartiles) during the first and third trimester of pregnancy, with the neuropsychological scores, were evaluated using multivariate linear regression analyses. The first exposure quartile was used as a reference category in these regression models. We explored the associations in both n-3 PUFA exposure periods (first and third trimesters of pregnancy), since in biological terms, it is interesting to investigate the potential associations in both periods of uterine neurodevelopment. However, we focused the presentation of the main findings on the ones observed in the first pregnancy trimester, due to the fact that they were found to be the most significant results.

All the regression models were adjusted for two sets of potential confounders. The minimally adjusted model used the most basic confounders, and the fully adjusted model used all the selected confounders. All the confounders were identified and selected before performing the analyses via the directed acyclic graph (DAG) model, illustrated in the Appendix A, Figure 1 [29], and a literature review for further selecting confounders [15,16]:(1)Minimally adjusted regression models included co-variate adjustments for maternal energy intake (kcal/day), sex of the child, age of the child at the time of testing, cohort location, and quality of the test.(2)Fully adjusted regression models additionally included co-variate adjustments for the birthweight of the child, gestational age, the duration of breastfeeding, maternal alcohol consumption, maternal education, maternal smoking, maternal social class, and maternal pre-pregnancy BMI, as well as number of pregnancies before the index pregnancy, number of miscarriages, and maternal country origin (Spain, or other).

The main tables of this paper present the findings of the n-3 PUFA intake in the first trimester of pregnancy, and additional statistical analyses were conducted to further explain the main results. Firstly, in order to remove any potential selection bias from the variables in the fully adjusted model, given the loss of observations, inverse probability weighting was applied [30]. Secondly, to evaluate any percentage change from the covariates included in the regression models, a change-in-estimates method was used. This was performed with a stepwise approach that allowed the identification of any changes to the effect estimate. However, a prior selection of the main confounders was performed by applying a DAG model, as mentioned before.

An extended list of co-variable adjustments, beyond the set previously selected, were performed as secondary analyses. Finally, the descriptive and regression analyses were repeated using n-3 PUFA intake in the third trimester of pregnancy as the exposure variable in secondary analyses, and the results are mostly presented in the Appendix A.

All the analyses were conducted with the Stata Statistical Software: Release 16. College Station, TX: StataCorp LLC, and statistical significance was set at *p* < 0.05.

## 3. Results

The mean dietary n-3 PUFA intakes of all pregnant mothers in their first and third trimesters were 1.49 g/d (SD = 0.35) and 1.44 g/d (SD = 0.34), respectively. Table 1 shows each family’s socio-demographic characteristics according to maternal n-3 PUFA intake during the first trimester of pregnancy. The n-3 PUFA intake was higher amongst women who were older, had a higher pre-pregnancy BMI, were born in Spain, and had a higher level of education and social class. Fully extended descriptive tables for both the first and third trimesters are shown in the Appendix A. Both trimesters showed similar patterns, with higher n-3 PUFA intakes in those mothers with more social advantages and healthier lifestyles. A small subsample (*n* = 948) of cord blood levels of fatty acids (omega-6/omega-3 ratio) showed moderate differences by maternal n-3 PUFA intake quartiles in the third pregnancy trimester. Finally, cord mercury levels were positively correlated to n-3 PUFA intake in both trimesters (Appendix A).

Neuropsychological test scores showed moderate differences between maternal n-3 PUFA intake during the first trimester of pregnancy (see Table 2). The MCSA outcomes presented a higher mean score per each increasing exposure quartile group. The ANT–HRT-SE outcome showed the same pattern, but with a lower mean score per increasing exposure quartile; however, the higher the HRT-SE score, the lower the attention performance. The Bayley Scale scores showed no differences for any of the exposure groups from the first and third pregnancy trimesters (Appendix A), and the other neuropsychological outcomes showed weaker mean score differences between the exposure quartiles, including the third pregnancy trimester, are shown in the Appendix A.

Table 3 shows minimally and fully adjusted associations between maternal n-3 PUFA consumption in the first trimester and child neuropsychological test results from MSCA and ANT using multivariable regression analyses. In the fully adjusted models, results were similar to those found in the minimally adjusted models, with significant associations between n-3 PUFA intake (fourth quartile compared to first quartile) and higher scores were observed in MSCA–General Cognitive (β = 2.26, 95% CI: 0.41, 4.11), MSCA–Verbal (β = 2.48, 95% CI: 0.53, 4.43), MSCA–Executive Function (β = 2.06, 95% CI: 0.17, 3.95), and ANT–HRT-SE in milliseconds (β = −11.52, 95% CI:−22.95, −0.09). Further adjustments for maternal verbal IQ proxy, maternal mental health, using supplements during pregnancy, pregnancy aMED, and cord mercury concentration, did not change the results (data not shown). Appendix A shows the non-significant associations found in an extended list of cognitive outcomes, including first year Bayley Scales. Appendix A shows associations between n-3 PUFA intake during the third trimester of pregnancy and neuropsychological test scores; the findings show much weaker associations than those observed in Table 3.

Repeating the fully adjusted models of Table 3 and Appendix A, Figure 2 shows the association coefficients of both n-3 PUFA intake trimesters (first and third). In both cases, we compared the fourth quartile group with the first quartile group of the exposure, and they showed a clear difference between the two intake periods, with larger coefficients for the associations between the first pregnancy trimester n-3 PUFA intakes and the main outcomes.

The findings shown in Table 3 were similar after applying the inverse probably weighing correction (Table 4). This means that there were no differences in the associations due to the missing cases. Furthermore, in Table 5, we observed that maternal social class and maternal education were the main confounders in the association between the exposure and the outcome, since they showed the higher percentage changes in the coefficient of the independent variable in the final model. This pattern was repeated in each of the main outcome models. Finally, repeating the models of Table 3 and treating the exposure as a continuous variable presented similar results (data not shown).

## 4. Discussion

In this population-based birth cohort study, some associations between maternal dietary n-3 PUFA intake during the first trimester of pregnancy and child neuropsychological scores at ages 4 and 7 years were found. The data suggest that the n-3 PUFA intakes in the third trimester of pregnancy seemed to be less related to later child neuropsychological function. This finding shows a discrepancy with the established hypothesis that most of the n-3 PUFAs are transferred from mother to child in the third pregnancy trimester [8]. Young children’s general cognition, including executive and verbal functions, and children’s attention functioning, were positively associated with maternal intakes of these fatty acids during early pregnancy. However, this fact does not suggest causality in the association, and large randomized clinical trials are needed. Furthermore, the study results were similar after applying inverse probability corrections for the missing cases. Finally, as expected, maternal social class and education level were the most important confounders in the exposure–outcome associations reported here.

The positive associations found here were also observed in other longitudinal cohort studies based on PUFA intakes (or similar nutrient compounds, such as seafood) during pregnancy and child neuropsychological development. For example, Mendez et al., 2009, in our previous study with 482 pregnant women from Minorca island, found that mothers who consumed two to three servings of fish per week during pregnancy had children with significantly higher scores in neuropsychological tests at 4 years of age, compared to mothers who had less than one serving of fish [31]. In this study, a statistical difference was found in all main McCarthy outcomes (general cognitive, perceptual performance, memory, verbal, numeric, and motor skills). Furthermore, a review paper from Weiser et al., 2016, described several studies about DHA intake during pregnancy and infancy, and showed positive associations similar to those seen in this study, for example, in child cognitive functions related to attention, memory, and verbal scores [9]. This review included a large cohort study report, including one by Hibbeln et al., 2007, who used pregnancy FFQs and child neuropsychological outcomes up until 8 years of age, and who discovered an association between a low intake of seafood, defined as less than 340 g per week during the third trimester of pregnancy, and an increased risk of suboptimal neuropsychological outcomes, affecting verbal function, fine motor, and social development scores [32]. These studies did not estimate the PUFA intake from FFQ, and used primary source compounds, such as fish intake, and neither assessed the exposure (PUFA intake) several times during pregnancy. Furthermore, there are a few randomized trials exploring the effect of pregnancy PUFA intake on later child neuropsychological development with positive results [8]. For example, in a randomized double-blinded clinical trial including 590 pregnant women, led by Helland in 2003, it was reported that daily n-3 PUFA supplementation during pregnancy had greater effects on mental processing outcomes at 4 years of age [33], which may be a similar outcome to our study’s ANT–HRT-SE assessed at 8 years of age.

In relation to the neuropsychological outcomes mostly related to maternal n-3 PUFA intake, it is interesting to see that executive and attention functions were highly associated. These functions are involved in the optimal prefrontal cortex development, the section of the brain that performs several complex cognitive functions, including the functions assessed in this study [34]. This brain area may need high amounts of n-3 PUFA during development due to the complexity of its synaptic connections.

Therefore, a mother’s n-3 PUFA intake during pregnancy may be vital for optimal long-term neuropsychological outcomes for the child. There are biological pathways facilitating the transfer between the mother and the offspring during pregnancy. One study from Koletzko et al., 2007, demonstrated an active and preferential maternal–foetal transfer of DHA across the human placenta, and that this pathway was a mechanism for the expression of human placental fatty acid binding and transport proteins [35]. Our results may be partly explained by this active transfer process of n-3 PUFAs from mother to foetus, and by the metabolic pathways involved, which are essential for the rapid cellular uptake of the n-3 PUFAs. Thus, both biological mechanisms may affect neurodevelopment during foetal growth [7,8,9,13]. For example, in our recent study conducted with this cohort (Julvez et al., 2020), we demonstrated the importance of some single nucleotide polymorphism (SNP) metabolizers of n-3 PUFAs, present in relation to maternal seafood intake, and later child attention function [36]. We further described stronger associations between early pregnancy n-3 PUFA intake and child neuropsychological development. We would have expected stronger associations at the end of the pregnancy, due to the fact that, in this period, neuron dendritic growth and myelination are highly activated [8]. However, the complexity of human brain growth during the entire gestational period and the related long-term behavioural consequences are difficult to assess. This fact requires more scientific work and a deeper analysis of the potential biological mechanisms. These finding needs to be confirmed in other epidemiological studies and further investigated in experimental studies, particularly in experiments focusing on the human brain development.

The main strengths of this study include its longitudinal cohort study design, with a large sample size from different regions of Spain, where all of the cohorts used common assessment protocols. The data in this study were collected prospectively and with standardized and valid methods and instruments regarding the exposure variables, covariates, and neuropsychological outcomes. For example, the neuropsychological assessments used validated and standardized tests, and the psychologists involved in the study underwent training and followed quality controls. The study also applied several statistical analyses in order to verify that the associations reported were independent to the main confounders, and also that the associations were not biased due to missing data.

The study design is not without limitations. Firstly, self-reported data from FFQ are subject to measurement errors and recall bias. The accuracy of the FFQ relies on the subject, which poses risks relating to the true representation of intake, leading to underestimation or overestimation. Furthermore, the conversion of results from the FFQ to nutrients are estimates at risk for variances, especially for foods with limited nutritional information, such as ready-to-eat meals prepared outside the home. Secondly, although a lot of covariate information was available, the study lacked dietary and supplementary information and biomarkers of n-3 PUFA intake during lactation, which may have been relevant for analysis. Additionally, although confounding factors, based on a literature review and a DAG model, were reviewed during data analysis, there is still a risk of potential residual confounders due to the observational nature of the study, causing potential bias in the study. Furthermore, the data showing the important confounding effect of maternal social class are indicative of this risk of potential residual confounding due to socially advantageous families. Furthermore, complete information for those that were lost or opted out are not available from all cohorts, creating possible selection bias. Nevertheless, we applied inverse probably weighting corrections in our study, with no changes observed in the main findings. However, some cohorts had certain information, such as low socioeconomic status, that were more prevalent amongst non-participants [17], and this has to be taken into consideration if results are generalized to the general population. A slight correlation between omega-3 PUFA intakes during the third trimester of pregnancy and cord blood omega-3 fatty acid concentration was observed in a subsample, and this low biomarker correlation is usually observed in this type of exposure based on FFQs [20,21], and plasma PUFA biomarkers are used to indicate short-term—about a few weeks—PUFA consumption [20]. Furthermore, a few women (5%) reported using omega-3 supplements during pregnancy, and we treated this as a separate variable, but we observed no influence in the present results. Finally, the clinical interpretation of the findings is limited by the fact that the associations were modest, since only a few test scores (MCSA points or ANT–HRT-SE milliseconds) separated the differences between exposure groups.

Overall, this study found that higher maternal n-3 PUFA intake during the first trimester of pregnancy was associated with improved scores in some child neuropsychological outcomes at 4 years and 7 years of age. The associations were only found in maternal n-3 PUFA intake during the first pregnancy trimester compared with the third trimester period. Given the large number of potential covariates, the temporal distance between exposure and outcome, and the non-clinical significance of the associations found, this research topic needs more longitudinal cohort studies to continue exploring these associations, plus clinical and experimental studies to further explain the biological pathways of n-3 PUFAs during pregnancy and their potential role in child neuropsychological development.

## Figures and Tables

**Figure 1 nutrients-14-00518-f001:**
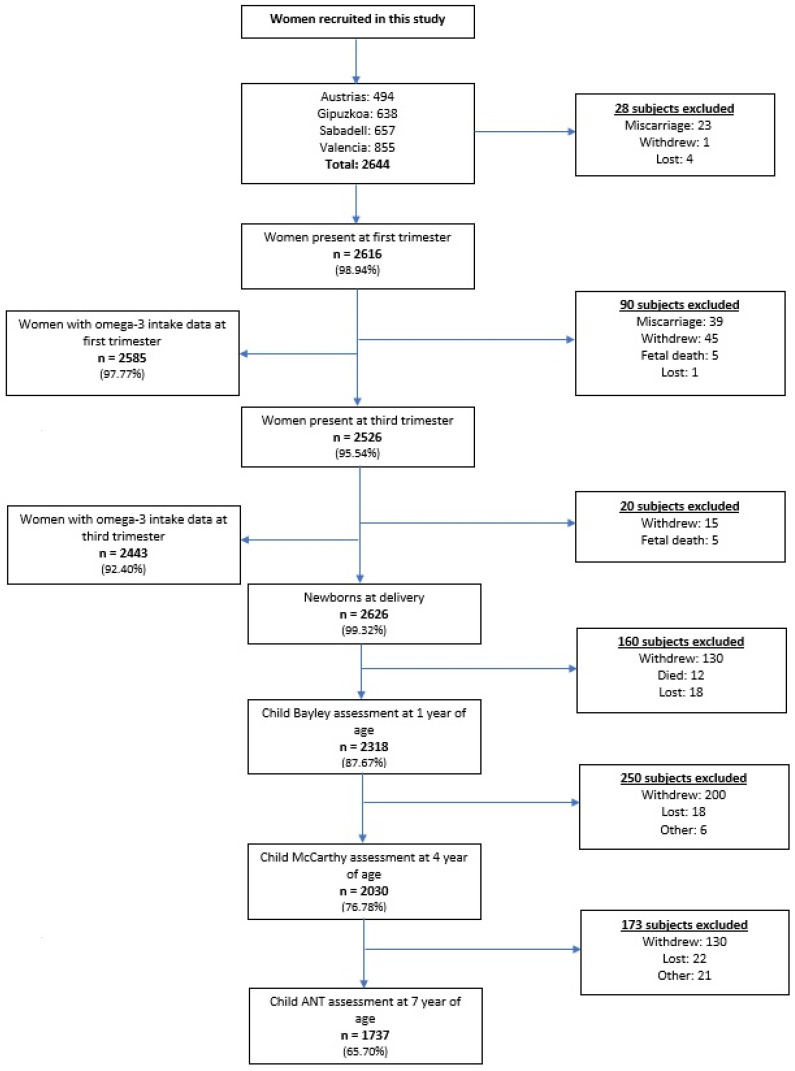
Flowchart of study population from Spanish Childhood and Environment (INMA) project, 2004–2016. Note: Numbers of excluded may not add up to next total number due to likely missing data or incomplete test. All percentages in brackets are portions of number of subjects relevant to the 2644 recruited women.

**Figure 2 nutrients-14-00518-f002:**
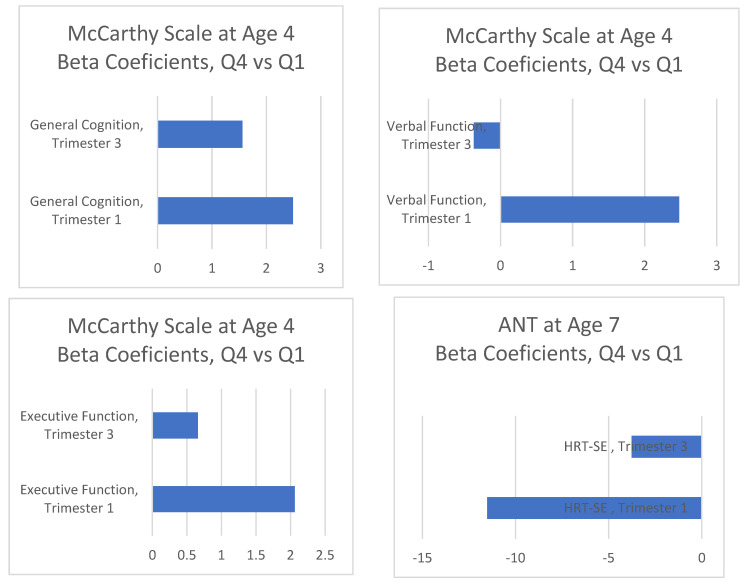
Fully adjusted regression coefficients ^a^ of the n-3 PUFA intakes during the first and third pregnancy trimesters and the main child neuropsychological outcomes: comparison of the associations of the two intake periods. (^a^ Multivariate linear regression models adjusted for sex of child, child’s age at testing, cohort, quality of the test (none for ANT), maternal energy intake, birthweight of the child, gestational age, duration of breastfeeding, maternal alcohol consumption, maternal education, maternal smoking, maternal social class, pre-pregnancy BMI, number of pregnancies, number of miscarriages, and maternal country of origin).

**Table 1 nutrients-14-00518-t001:** Baseline characteristics of the study participants according to the quartiles of dietary omega-3 fatty acids consumption in the first trimester of pregnancy.

	Quartiles of Dietary Omega ^b^ 3 Intake at 1st Trimester *				*p* Values
	1	2	3	4	
	(*n* = 645) ^a^	(*n* = 645) ^a^	(*n* = 645) ^a^	(*n* = 647) ^a^	
Age (years) ^c^, mean (SD)	31.32 (4.36)	31.84 (4.33)	32.25 (3.97)	32.37 (4.24)	0.001
Mother education level, *n* (%)					0.131
Primary or less	177 (27.48)	169 (26.24)	152 (23.49)	150 (23.22)	
Secondary school	264 (40.99)	248 (38.51)	263 (40.65)	291 (45.05	
University or more	203 (31.52)	227 (35.25)	232 (35.86)	205 (31.73)	
Social class of mother, *n* (%)					0.115
Highly skilled	123 (19.07)	142 (22.02)	143 (22.07)	144 (22.29)	
Non-manual	152 (23.57)	158 (24.50)	186 (28.70)	168 (26.01)	
Manual	370 (57.36)	345 (53.49)	319 (49.23)	334 (51.70)	
Smoking during pregnancy, *n* (%)					0.783
Yes	101 (16.67)	98 (16.25)	102 (16.64)	112 (18.30)	
No	505 (83.33)	505 (83.75)	511 (83.36)	500 (81.70)	
Alcohol consumption during first trimester, *n* (%)					0.048
Yes	172 (26.67)	197 (30.54)	219 (33.80)	201 (31.07)	
No	473 (73.33)	448 (69.46)	429 (66.20)	446 (68.93)	
Energy intake (kcals/day), mean (SD)	2115.48 (570.06)	2147.30 (559.07)	2127.43 (572.07)	2089.87 (523.77)	0.308
Preterm, *n* (%)					0.836
Yes	32 (5.24)	26 (4.25)	27 (4.33)	29 (4.73)	
No	579 (94.76)	586 (95.75)	597 (95.67)	584 (95.27)	
Mother BMI pre-pregnancy, mean (SD)	23.51 (4.39)	23.46 (4.27)	23.29 (3.86)	24.04 (4.70)	0.012
Number of pregnancies, *n* (%)					0.001
1	308 (47.75)	305 (47.29)	291 (44.91)	272 (42.04)	
2	226 (35.04)	246 (38.14)	224 (34.57)	214 (33.08)	
3 or more	111 (17.21)	94 (14.57)	133 (20.52)	161 (24.88)	
Number of previous miscarriages, *n* (%)					0.151
0	492 (76.28)	510 (79.07)	499 (77.13)	477 (73.72)	
1 or more	153 (23.72)	135 (20.93)	148 (22.87)	170 (26.28)	
Country of birth of mother, *n* (%)					0.001
Spain	550 (85.27)	588 (91.30)	614 (94.75)	612 (94.88)	
Other	95 (14.73)	56 (8.70)	34 (5.25)	33 (5.12)	
Cohort location, *n* (%)					0.001
Asturias	103 (15.97)	95 (14.73)	115 (17.75)	169 (26.12)	
Gipuzkoa	183 (28.37)	170 (26.36)	155 (23.92)	119 (18.39)	
Sabadell	123 (19.07)	149 (23.10)	187 (28.86)	195 (30.14)	
Valencia	236 (36.59)	231 (35.81)	191 (29.48)	164 (25.35)	
Cord blood omega 6(AA)/omega 3 (EPA + DHA) ratio, mean (SD)	2.98 (0.83)	2.94 (0.89)	2.93 (0.84)	2.87 (0.77)	0.560
Breastfeeding (weeks), *n* (%)					0.194
0	87 (15.48)	77 (16.96)	97 (16.96	79 (13.64)	
>0–16	157 (27.94)	156 (26.94)	129 (22.55)	140 (24.18)	
>16–24	82 (14.59)	103 (17.79)	92 (16.08)	86 (14.85)	
>24	236 (41.99)	243 (41.97)	254 (44.41)	274 (47.32)	
Child Characteristics					
Sex, *n* (%)					0.459
Female	300 (48.78)	300 (48.47)	317 (50.48)	285 (45.97)	
Male	315 (51.22)	319 (51.53)	311 (49.52)	335 (54.03)	
Birthweight, *n* (%)					0.290
<3000 g	171 (27.94)	178 (28.94)	183 (29.33)	154 (25.00)	
3000–3500 g	263 (42.97)	258 (41.95)	269 (43.11)	299 (48.54)	
>3500 g	178 (29.08)	179 (29.11)	172 (27.56)	163 (26.46)	

* 1st quartile: <1.262 g/day; 2nd quartile: 1.262 g–1.440 g/day; 3rd quartile: 1.440 g–1.657 g/day; 4th quartile: >1.657 g/day. ^a^ Some categories may be missing values, and therefore totals may not match the total number of subjects. ^b^ Adjusted for energy intake. ^c^ Maternal age at delivery.

**Table 2 nutrients-14-00518-t002:** Mean test scores and standard deviation (SD) from McCarthy Scales of Children’s Abilities (MSCA) at 4 year of age, and Attention Network Test (ANT) at 7 years of age, according to maternal omega-3 intake frequency during the first trimester of pregnancy.

	Quartiles of Dietary Omega-3 ^b^ Intake at 1st Trimester				*p* Values
	1	2	3	4	
MSCA	*n* = 422	*n* = 444	*n* = 468	*n* = 461	
General Cognitive ^a^	98.59 (14.23)	100.45 (15.11)	100.70 (14.12)	101.11 (15.57)	0.060
Verbal ^a^	98.71 (14.19)	99.69 (14.81)	100.74 (14.70)	101.24 (15.72)	0.056
Executive Function ^a^	98.60 (14.48)	100.14 (15.17)	100.98 (14.33)	101.08 (15.24)	0.049
ANT	*n* = 379	*n* = 411	*n* = 444	*n* = 421	
HRT-SE ^c^	314.93 (80.45)	306.11 (83.50)	306.39 (81.24)	297.31 (83.57)	0.027

^a^ Index score centred to 100 with standard deviation of 15. ^b^ Adjusted for energy intake. ^c^ Score is inversed; higher score equals lower performance.

**Table 3 nutrients-14-00518-t003:** Multivariable regression analysis: associations between maternal omega-3 consumption in the first trimester of pregnancy and child’s scores on McCarthy Scales of Children’s Abilities (MSCA) at 4 years of age and Attention Network Test of Children’s Abilities (ANT) at 7 years of age.

Neuropsychological Outcome		No. of Subjects	Difference in Child’s Neuropsychological Score					
			Minimally adjusted ^a^			Fully adjusted ^b^		
			β	(95% CI)	*p* *	β	(95% CI)	*p* *
MSCA								
General Cognitive								
	First quartile	409	ref			ref		
	Second quartile	425	1.72	(−0.16, 3.60)	0.070	1.16	(−0.67, 2.98)	0.212
	Third quartile	443	2.19	(0.32, 4.05)	0.020	1.61	(−0.21 3.43)	0.084
	Fourth quartile	436	2.64	(0.76, 4.52)	0.006	2.26	(0.41, 4.11)	0.017
Verbal								
	First quartile	409	ref			ref		
	Second quartile	425	0.82	(−1.12, 2.77)	0.405	0.35	(−1.57, 2.27)	0.722
	Third quartile	443	2.16	(0.24, 4.10)	0.028	1.82	(−0.10, 3.7)	0.063
	Fourth quartile	436	2.61	(0.66, 4.55)	0.009	2.48	(0.53, 4.43)	0.013
Executive Function								
	First quartile	409	ref			ref		
	Second quartile	425	1.39	(−0.52, 3.31)	0.153	1.00	(−0.86, 2.87)	0.291
	Third quartile	443	2.43	(0.53, 4.33)	0.012	1.88	(0.01, 3.74)	0.049
	Fourth quartile	436	2.55	(0.63, 4.47)	0.009	2.06	(0.17, 3.95)	0.033
ANT								
HRT-SE								
	First quartile	363	ref			ref		
	Second quartile	392	−8.22	(−19.14, 2.68)	0.130	−7.29	(−18.50, 3.91)	0.200
	Third quartile	421	−9.01	(−19.77, 1.75)	0.100	−8.01	(−19.10, 3.09)	0.160
	Fourth quartile	398	−12.42	(−23.41, −1.43)	0.030	−11.52	(−22.95, −0.09)	0.048

* *p* for trend not listed, but for all models the value was below 0.001. ^a^ Multivariate linear regression models adjusted for sex of child, child’s age at testing, cohort, quality of the test (none for ANT), and maternal energy intake. ^b^ Multivariate linear regression models adjusted additionally for birthweight of the child, gestational age, duration of breastfeeding, maternal alcohol consumption, maternal education, maternal smoking, maternal social class, pre-pregnancy BMI, number of pregnancies, number of miscarriages, and maternal country of origin.

**Table 4 nutrients-14-00518-t004:** Associations between maternal omega-3 intake in the first trimester of pregnancy and child’s neuropsychological scores. Comparison between previous final models and models with inverse probability weighting (IPW).

Neuropsychological Outcome		No. of Subjects	Difference in Child’s Neuropsychological Score					
			Fully adjusted ^a^			Inverse probability weighting ^b^		
			β	(95% CI)	*p*	β	(95% CI)	*p*
MSCA		1713						
General Cognitive								
	First quartile		ref			ref		
	Second quartile		1.16	(−0.67, 2.98)	0.212	0.95	(−0.92, 2.81)	0.320
	Third quartile		1.61	(−0.21 3.43)	0.084	1.98	(0.19, 3.77)	0.031
	Fourth quartile		2.26	(0.41, 4.11)	0.017	2.49	(0.62, 4.36)	0.009
Verbal								
	First quartile		ref			ref		
	Second quartile		0.35	(−1.57, 2.27)	0.722	0.20	(−1.71, 2.10)	0.841
	Third quartile		1.82	(−0.10, 3.7)	0.063	2.08	(0.21, 3.94)	0.029
	Fourth quartile		2.48	(0.53, 4.43)	0.013	2.76	(0.83, 4.69)	0.005
Executive Function								
	First quartile		ref			ref		
	Second quartile		1.00	(−0.86, 2.87)	0.291	0.73	(−1.15, 2.62)	0.447
	Third quartile		1.88	(0.01, 3.74)	0.049	2.23	(0.38, 4.08)	0.019
	Fourth quartile		2.06	(0.166, 3.95)	0.033	2.54	(0.65, 4,43)	0.008
ANT								
HRT-SE		1574						
	First quartile		ref			ref		
	Second quartile		−7.29	(−18.50, 3.91)	0.200	−8.34	(−23.66, 1.25)	0.078
	Third quartile		−8.01	(−19.10, 3.09)	0.160	−8.90	(−20.53, 3.06)	0.147
	Fourth quartile		−11.52	(−22.95, −0.09)	0.048	−14.34	(−30.76, −6.12)	0.003

^a^ Multivariate linear regression models adjusted for sex of child, child’s age at testing, cohort, quality of the test (none for ANT and N-back), maternal energy intake, birthweight of the child, gestational age, duration of breastfeeding, maternal alcohol consumption, maternal education, maternal smoking, maternal social class, pre-pregnancy BMI, number of pregnancies, number of miscarriages, and maternal country of origin, ^b^ Results were additionally IPW corrected by child’s age at testing, cohort, maternal energy intake, maternal education, maternal smoking, maternal social class, and quality of the test (none for ANT and N-back).

**Table 5 nutrients-14-00518-t005:** Percentage change by the confounding variables of the associations between maternal omega-3 intake in the first trimester of pregnancy and child’s neuropsychological scores.

Neuropsychological Outcome	Difference in Child’s Neuropsychological Score	
	Minimally adjusted ^a^	Change in %
	β (95% CI)	
MSCA		
Executive Function		
Minimially adjusted model	2.15 (0.15, 4.14)	
Maternal social class	1.78 (−0.18, 3.74)	−17.06
Maternal education	1.96 (0.03, 3.89)	10.12
Pre-pregnancy BMI	2.06 (0.12, 3.99)	4.93
Child’s birth weight	2.03 (0.10, 3.96)	−1.37
Alcohol consumption	2.02 (0.08, 3.95)	−0.61
Smoking during pregnancy	2.02 (0.09, 3.95)	0.18
General Cognitive		
Minimially adjusted model	2.12 (0.17, 4.07)	
Maternal social class	1.72 (−0.18, 3.62)	−18.76
Maternal education	1.90 (0.03, 3.78)	10.4
Pre-pregnancy BMI	1.98 (0.10, 3.85)	3.91
Child’s birth weight	1.95 (0.07, 3.82)	−1.44
Alcohol consumption	1.92 (0.05, 3.80)	−1.35
Smoking during pregnancy	1.93 (0.05, 3.80)	0.24
Verbal		
Minimially adjusted model	2.14 (0.11, 4.17)	
Maternal social class	1.82 (−0.18, 3.83)	−14.85
Maternal education	1.99 (0.01, 3.97)	9.14
Pre-pregnancy BMI	2.11 (0.13, 4.09)	5.99
Child’s birth weight	2.10 (0.12, 4.08)	−0.48
Alcohol consumption	2.10 (0.12, 4.09)	0.19
Smoking during pregnancy	2.10 (0.12, 4.08)	−0.1
ANT		
HRT-SE		
Minimally adjusted model	−8.50 (−20.05, 3.05)	
Maternal social class	−7.45 (−18.97, 4.08)	−12.41
Maternal education	−7.79 (−19.29, 3.71)	4.58
Pre-pregnancy BMI	−7.98 (−19.50, 3.54)	2.44
Child’s birth weight	−8.02 (−19.55, 3.50)	0.57
Alcohol consumption	−8.01 (−19.51, 3.50)	−0.23
Smoking during pregnancy	−7.99 (−19.51, 3.53)	−0.17

^a^ Multivariate linear regression minimally adjusted models adjusted for sex of child, child’s age at testing, cohort, quality of the test (none for ANT and N-back), and maternal energy intake.

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
