# Peer review of "Omega-3 Fatty Acid Intake during Pregnancy and Child Neuropsychological Development: A Multi-Centre Population-Based Birth Cohort Study in Spain"

_nutrients, 2022, doi:10.3390/nu14030518_

Round 1

Reviewer 1 Report

The authors have carried out an epidemiology study to determine the cognitive benefits of O3.  They enrolled multiple cohorts and combined the data and report finding associations with 1st trimester O3s and children’s neurodevelopmental scores at 4 and 7 years of age.  They conclude that O3s do benefit children’s cognition, which matches the opinion of many scientists even though the evidence supporting it is thin. 

The study has a large n obtained by enrolling participants at four sites. Observational epidemiology studies are complex as illustrated by the DAG in the supplementary material. This paper combines sophisticated measures (MSCA, ANT) and unsophisticated ones (O3 values determined from FFQ, BSID 1 [there are 2 subsequent versions]. It also has a complex analysis plan with an unspecified number of analyses. Given the time span between the mothers first trimester and the child being 8 year many intervening events can easily influence the results.  These factors make a definite conclusion questionable at best. 

Epidemiology studies find associations.  They do not prove causal relationships, but such relationships seem likely when different investigators using the same tests obtain the same results. Consider revising some of the language about the findings to be more realistic (millisecond differences between O3 exposures in the first trimester and outcomes at 8 years would have no clinical significance and seem most unlikely).   Consider clarifying how the study handled uniform testing and scoring between the centers. Were there other tests that the children were given besides the 3 reported on?  Were clinical testers blind to the exposure of interest?  Please clarify the flow of the analyses and which results the authors are reporting. There appear to be two models (full and reduced) for each O3 exposure (1st and 3rd trimester).  Were the models determined before the analysis?

Specifics:

Title       The study is of O3’s. The authors use a FFQ to determine O3 exposure, but do not talk about the foods consumed. Consider revising the title.

Abstract:

Line 57                 Please include BSID 1 (there are 3 versions)

                Line 59                 The conclusions support the prevailing belief that LCP, especially DHA & EPA are important for brain development (11.52 milliseconds on the ANT is miniscule under any circumstances).

Introduction:

                Line 102               Reference 3 is not available online

                Line 105               Synthesis of DHA from ALA in humans is at a very low level (~1% of what the brain needs so DHA must be acquired preformed). 

Study Population:

                Line 152                Did all 4 sites use the same questionnaires and data collection procedures?

Line 160               Were there no exclusions for birth defects, gestational age, maternal illnesses, etc.?

Exposure Information:

                Line 179               Reference 18 contains no mention of LCP in the validation study. The FFQ was validated for a number of nutrients, etc., but the reference has no mention of O3.

Co-variable information:

                Line 201               Did the authors really examine these multiple other exposures?  If so, it increases the number of analyses and thus weakens the study.  Given enough variables and analyses one is bound to find associations. 

                Line 214                 Please revise to clarify this was the mother’s VIQ.

Statistical Analysis:          It appears there are primary and secondary analyses and full and reduced models (uniform labeling would help the reader to follow what the authors did).  As currently written it is difficult to keep all the models straight. I suggest each have their own paragraph.  Did the analyses include stepwise regression?  If so, are the results from the full or reduced model before eliminating covariates?

                Line 245                Please clarify why there is a difference between quartiles for LCP by trimesters.

                Line 252               Please spell out Chi Squared test.  

                Line 266                If there were nationalities apart from Spanish please clarify.

                Line 272               Does this sentence refer to stepwise regression?

Results:                This section would be easier to follow if organized by primary (? first trimester O3s) & secondary (? third trimester O3s) analyses and full and reduced models. Were the analyses predetermined? Since the majority of O3 are transferred to the fetus in the third trimester the authors should justify why they selected the first trimester for the primary analysis?

Discussion:

                Line 350                The authors should mention this is their own study and not a different group of independent investigators.

                Line 355                I suggest the authors cite the paper they are referencing rather than cite a review from within a review.   

                Line 386               grammar

                Lines 393-6          grammar and meaning

                Line 398                It really isn’t a validated tool for O3 unless there is another reference.  

Author Response

Reviewer-1

The authors have carried out an epidemiology study to determine the cognitive benefits of O3.  They enrolled multiple cohorts and combined the data and report finding associations with 1st trimester O3s and children’s neurodevelopmental scores at 4 and 7 years of age.  They conclude that O3s do benefit children’s cognition, which matches the opinion of many scientists even though the evidence supporting it is thin. 

Response to the Reviewer: The Reviewer is right in these statements, except that we combined different cohort studies. This is not what we did. INMA study is a multi-center birth cohort study, and each site forms part of the same study following prospectively the same protocol.

We have clarified it better in Methods section, see changes in lines 156-161.

The study has a large n obtained by enrolling participants at four sites. Observational epidemiology studies are complex as illustrated by the DAG in the supplementary material. This paper combines sophisticated measures (MSCA, ANT) and unsophisticated ones (O3 values determined from FFQ, BSID 1 [there are 2 subsequent versions]. It also has a complex analysis plan with an unspecified number of analyses. Given the time span between the mothers first trimester and the child being 8 year many intervening events can easily influence the results.  These factors make a definite conclusion questionable at best. 

Response to the Reviewer: Thanks for this comment, the Reviewer is right, we need to be careful in using strong statements in the conclusion of this study.

We have changed the conclusions accordingly. See changes in Abstract section, lines 69-72. See changes in Discussion section, lines 496-505.

Epidemiology studies find associations.  They do not prove causal relationships, but such relationships seem likely when different investigators using the same tests obtain the same results. Consider revising some of the language about the findings to be more realistic (millisecond differences between O3 exposures in the first trimester and outcomes at 8 years would have no clinical significance and seem most unlikely).   

Response to the Reviewer: We agree with the Reviewer’s comment, we need to be careful in the interpretation of the results in epidemiological studies. We have moderated the language in the interpretation of the results, we also have included some of the limitations commented by the Reviewer.

Counting on the changes added in previous comment, we also made changes in the next lines of the discussion section: 386-397, 483-486.

Consider clarifying how the study handled uniform testing and scoring between the centers. Were there other tests that the children were given besides the 3 reported on?  Were clinical testers blind to the exposure of interest? 

Response to the Reviewer: Thank you for this comment, we added these clarifications in the Methods section, lines 240-244.

Please clarify the flow of the analyses and which results the authors are reporting. There appear to be two models (full and reduced) for each O3 exposure (1st and 3rd trimester).  Were the models determined before the analysis?

Response to the Reviewer: Thank you for this comment, yes, the Reviewer is right, we used two models and the models were determined before performing the analyses. We have better clarified this in the Methods section.

Please see these clarifications in Methods section, lines 288-298.

Specifics:

Title       The study is of O3’s. The authors use a FFQ to determine O3 exposure, but do not talk about the foods consumed. Consider revising the title.

Response to the Reviewer: Ok, title changed accordingly.

Abstract:

Line 57                 Please include BSID 1 (there are 3 versions)

Response to the Reviewer: Ok, text changed accordingly. Line 58.

Line 59                 The conclusions support the prevailing belief that LCP, especially DHA & EPA are important for brain development (11.52 milliseconds on the ANT is miniscule under any circumstances).

Response to the Reviewer: Ok, text changed accordingly. Lines 70-72.

Introduction:

 Line 102               Reference 3 is not available online

Response to the Reviewer: Ok, Reference 3 updated. Line 104.

 Line 105               Synthesis of DHA from ALA in humans is at a very low level (~1% of what the brain needs so DHA must be acquired preformed). 

Response to the Reviewer: Ok, we have updated the sentences accordingly, however, in pregnancy, it seems to be increased this rate. Lines 111-112.

Study Population:

                Line 152                Did all 4 sites use the same questionnaires and data collection procedures?

Response to the Reviewer: Yes, all the 4 sites used the same questionnaires and data collection protocol. We have added this clarification to the text in lines 158-160.

Line 160               Were there no exclusions for birth defects, gestational age, maternal illnesses, etc.?

Response to the Reviewer: No exclusions were based in these parameters, only no born or death at birth. However, this is a population-based birth cohort, and the a few cases of children with neurological problems or birth defects, they were then lost to follow up and no cognitive testing was applied. Children born with low gestational age or with maternal illnesses during pregnancy were included. But they are a few and excluding them did not affect the findings.

Exposure Information:

                Line 179               Reference 18 contains no mention of LCP in the validation study. The FFQ was validated for a number of nutrients, etc., but the reference has no mention of O3.

Response to the Reviewer: The reproducibility and validity of the FFQ was, in general, acceptable for most nutrients and food group intakes, similarly to the observed in literature. Regarding the intake of omega 3 (O3), our FFQ showed a good reproducibility for the intake of omega 3, correlation coefficient, r=0.50 (Vioque et al, 2013). In a posterior study (Chisaguano et al, 2014), when the intake of O3 measured by the FFQ was compared with the O3 plasma concentrations in a group of 496 pregnant women of the INMA study, the coefficient correlation was lower than the correlation for reproducibility although statistically significant (r=0.19, p<0.001). Lower correlations are usually found in the literature when comparing intakes with plasma concentrations (Willett, 1998).

According to this point we propose to include a new paragraph and reference in the manuscript, Methods section, lines 204-214.

New reference: Chisaguano AM, Montes R, Castellote AI, Morales E, Júlvez J, Vioque J, Sunyer J, López-Sabater MC. Elaidic, vaccenic, and rumenic acid status during pregnancy: association with maternal plasmatic LC-PUFAs and atopic manifestations in infants. full text links Pediatr Res 2014; 76(5): 470-6.  doi: 10.1038/pr.2014.119

Co-variable information:

Line 201               Did the authors really examine these multiple other exposures?  If so, it increases the number of analyses and thus weakens the study.  Given enough variables and analyses one is bound to find associations. 

Response to the Reviewer: Thanks for the comment, but in this section, we only comment of the data that was gathered in the questionnaires, but for the multivariate analyses, we previously selected some of them. But we wanted to show to the reader all the data that we gathered that would be important to at least consider, before pointing out which variables would be used in the main analyses.  We have added a clarification in Methods section, lines 218-219.

Line 214                 Please revise to clarify this was the mother’s VIQ.

Response to the Reviewer: Yes, it is. We have clarified in the line 236.

Statistical Analysis: It appears there are primary and secondary analyses and full and reduced models (uniform labeling would help the reader to follow what the authors did).  As currently written it is difficult to keep all the models straight. I suggest each have their own paragraph.  Did the analyses include stepwise regression?  If so, are the results from the full or reduced model before eliminating covariates?

Response to the Reviewer: Thanks for the comments and input in this section. We have added changes and clarifications in order to be easier to follow for the reader. In response to the use of stepwise method, we only used it in order to perform a complementary analysis shown in the Results in Table 5. But not for selecting the confounders for the main multi-variate analyses, in this case we used a DAG model. I hope that this is now clearer with the updated modifications in the text. See changes in Methods section, lines 288-317.

Line 245                Please clarify why there is a difference between quartiles for LCP by trimesters.

Response to the Reviewer: Ok, we added this clarification in lines 277-279.

Line 252               Please spell out Chi Squared test.  

Response to the Reviewer: Ok, done. Lines 282-283.

Line 266                If there were nationalities apart from Spanish please clarify.

Response to the Reviewer: Yes, but since they were a few and there was a disparity of countries, we rather prefer to show it as Spain and others. This is now clarified in the section, line 305.

Line 272               Does this sentence refer to stepwise regression?

Response to the Reviewer: We understand the Reviewer confusion, this is now clarified in the previous comment. Lines 312-313.

Results:                This section would be easier to follow if organized by primary (? first trimester O3s) & secondary (? third trimester O3s) analyses and full and reduced models. Were the analyses predetermined? Since the majority of O3 are transferred to the fetus in the third trimester the authors should justify why they selected the first trimester for the primary analysis?

Response to the Reviewer: Thank you for this comment, as this was already clarified, that we were looking in both exposure periods at the same time without any preference. But it is true that some literature focusses more on the late pregnancy, however there is still scientific discussion ground to understand the exact pregnancy windows. Here we have updated the Methods section text and justified that we focused to show the main significant results of early pregnancy without excluding the results of the late pregnancy in supplementary material. We prefer to keep the results as they are presently shown. However, as requested by the second Reviewer, we have included a Figure 2 comparing the coefficients between the two pregnancy periods. This new figure has added a more balanced presentation of findings of the two pregnancy periods. See again the lines 288-291 in Methods section. See Figure 2 and lines 357-361 in Results section.

Discussion:

Line 350                The authors should mention this is their own study and not a different group of independent investigators.

Response to the Reviewer: Ok, we have changed the text accordingly. Lines 400-401.

Line 355                I suggest the authors cite the paper they are referencing rather than cite a review from within a review.   

Response to the Reviewer: Ok, we have updated it and added the original reference. Line 413.

                Line 386               grammar

Response to the Reviewer: Ok, we have changed the sentence and improving the grammar. Lines 442-445.

                Lines 393-6          grammar and meaning

Response to the Reviewer: Ok, we have rewritten the full section. See changes in lines 450-459.

Line 398                It really isn’t a validated tool for O3 unless there is another reference.  

Response to the Reviewer: Ok, it is partially validated, as we answered previously in another comment, however, we have deleted this sentence. See changes in line 461.

Reviewer 2 Report

I reviewed the manuscript by Tahaei et al. The study explores the effects of omega-3 intake during the different stages of pregnancy on neuropsychological development of the child. The authors evaluated food frequency questionnaire in 2644 pregnant women recruited from different parts of Spain between 2004-2008. The most positive correlations were found between omega-3 intake during the first trimester. Child neuropsychological status was assessed in children at 1, 4 and 7 years old. Overall, the manuscript tests a clear hypothesis and the methods are well described; however, there are some issues that need to be addressed.

  1. My major comment is that the manuscript can benefit from figures instead of just the Tables. It is difficult to extract the information from tables. For example, the authors can include figures showing associations between first trimester vs. third trimester omega-3 intake and neuropsychological parameters that they evaluated.
  2. Some statements are too strong and may not be true. For example, line 112 states that there is currently no recommendations for daily intake of omega-3 fatty acids. This should be clarified.
  3. The title of the article has a grammatical error. It should ‘fatty acid-rich foods’, not ‘fatty acids rich foods’.
  4. The authors spelled DHA and EPA wrong. DHA is not called docosahexaenoic ‘fatty’ acids, it is docosahexaenoic acid.
  5. Line 123, please explain what it means that developing brain is more susceptible to n-3 PUFA deficiency.
  6. Line 166, 378 and 381 have typos, it should be fetal not foetal.
  7. Line 175, grammatical error. Is it ‘a’ or ‘the’, cannot be both.
  8. Authors mention that some pregnant women started participating the study after the birth of their child. If that is the case, how was the diet assessed in those mothers?
  9. Table 1, birthweight: there seems to be a problem. Did they mean > 3500 g?
  10. The Supp Info shows that some women had omega-3 supplements. How do the authors explain the effects of supplementation on their study outcomes? Perhaps they can comment on this in the Discussion.
  11. Line 291, the authors mention “Finally, cord mercury levels were positively correlated by n-3 PUFA intake in both trimesters”. Is this a concern? Please discuss. 
  12. Line 310 typo ‘concentration’, line 315 grammatical error.
  13. The sentence in Line 371 is not clear. “These functions are involved in the optimal prefrontal cortex development, the section of the brain that serves several complex cognitive functions, including the previous ones.” What are the previous ones? This sentence needs to be clarified.
  14. What is the overall recommendation for pregnant women that can be inferred from their study?
  15. Line 376 the authors state that “There are biological pathways describing the transfer between the mother and the offspring during pregnancy”. This needs to be further clarified. What are those pathways, and are those fatty acids transferred to cord blood? Is there a preference of transfer rate between DHA and ARA or EPA? Are there any association studies exploring the transfer of those fatty acids to cord blood and any associations with neuropsychological effects in children?
  1. Line 385, the sentence has grammatical errors and it is unclear.

“However, the complexity of human brain growth during the entire gestational period and its behavioural consequences in a long term are difficult to predict and there is gown for further scientific discussions of the potential biologic mechanisms...”

Author Response

Reviewer-2

I reviewed the manuscript by Tahaei et al. The study explores the effects of omega-3 intake during the different stages of pregnancy on neuropsychological development of the child. The authors evaluated food frequency questionnaire in 2644 pregnant women recruited from different parts of Spain between 2004-2008. The most positive correlations were found between omega-3 intake during the first trimester. Child neuropsychological status was assessed in children at 1, 4 and 7 years old. Overall, the manuscript tests a clear hypothesis and the methods are well described; however, there are some issues that need to be addressed.

  1. My major comment is that the manuscript can benefit from figures instead of just the Tables. It is difficult to extract the information from tables. For example, the authors can include figures showing associations between first trimester vs. third trimester omega-3 intake and neuropsychological parameters that they evaluated.

Response to the Reviewer: Ok, we agree with the Reviewer and we have added a Figure 2 in the manuscript, we liked to add the association coefficient comparisons between the two exposure periods.

See changes in the Result section, lines 357-361 and the Figure 2 located in lines 1006-1016.

  1. Some statements are too strong and may not be true. For example, line 112 states that there is currently no recommendations for daily intake of omega-3 fatty acids. This should be clarified.

Response to the Reviewer: We agree with the Reviewer, we have changed the sentence accordingly. See line 114.

  1. The title of the article has a grammatical error. It should ‘fatty acid-rich foods’, not ‘fatty acids rich foods’.

Response to the Reviewer: We agree with the Reviewer, we have changed the title accordingly. Since the previous Reviewer also suggested not to use the word “foods”.

  1. The authors spelled DHA and EPA wrong. DHA is not called docosahexaenoic ‘fatty’ acids, it is docosahexaenoic acid.

Response to the Reviewer: We agree with the Reviewer, we have changed the sentence accordingly. Lines 106-107.

  1. Line 123, please explain what it means that developing brain is more susceptible to n-3 PUFA deficiency.

Response to the Reviewer: Ok, we have added an explanation in lines 126-129.

  1. Line 166, 378 and 381 have typos, it should be fetal not foetal.

Response to the Reviewer: Ok, changes added accordingly.

  1. Line 175, grammatical error. Is it ‘a’ or ‘the’, cannot be both.

Response to the Reviewer: Ok, changes added accordingly. Line 184.

  1. Authors mention that some pregnant women started participating the study after the birth of their child. If that is the case, how was the diet assessed in those mothers?

Response to the Reviewer: The Reviewer is right, this need a clarification. Those a few participants were finally excluded from data analyses, since they had missing data on FFQ during pregnancy. This is now clarified in the text, lines 176-177.

  1. Table 1, birthweight: there seems to be a problem. Did they mean > 3500 g?

Response to the Reviewer:  The Reviewer is right, error corrected.

  1. The Supp Info shows that some women had omega-3 supplements. How do the authors explain the effects of supplementation on their study outcomes? Perhaps they can comment on this in the Discussion.

Response to the Reviewer:  Thank you for this comment, we have added a sentence in discussion that a few women (5%) were using PUFA supplementation and we used this a separate variable that did not influence the results. See the changes of the Discussion section, lines 482-483.

  1. Line 291, the authors mention “Finally, cord mercury levels were positively correlated by n-3 PUFA intake in both trimesters”. Is this a concern? Please discuss. 

Response to the Reviewer:  Ok, thanks, this is a good point. We have added a discussion of about this concern. We indicated, we did not observe any change in the findings after adding the mercury level in the models. However, it is important to keep the indication of consumption of fish with low levels of mercury, in order to avoid any potential developmental neurotoxicity.

See changes in the discussion section, lines 487-492.

  1. Line 310 typo ‘concentration’, line 315 grammatical error.

Response to the Reviewer: Thanks, changes added to the text. Lines 352 and 362.

  1. The sentence in Line 371 is not clear. “These functions are involved in the optimal prefrontal cortex development, the section of the brain that serves several complex cognitive functions, including the previous ones.” What are the previous ones? This sentence needs to be clarified.

Response to the Reviewer: Thanks for the comment, I refer the cognitive functions assessed in our study. I have clarified it in Discussion section, lines 424-425.

  1. What is the overall recommendation for pregnant women that can be inferred from their study?

Response to the Reviewer: Thanks, this is a good point. Linking this with the previous comment about mercury, we added an overall recommendation to increase PUFA intakes during pregnancy, but in a safe way and avoid large predatory fish.

See the changes in discussion section, lines 492-495.

  1. Line 376 the authors state that “There are biological pathways describing the transfer between the mother and the offspring during pregnancy”. This needs to be further clarified. What are those pathways, and are those fatty acids transferred to cord blood? Is there a preference of transfer rate between DHA and ARA or EPA? Are there any association studies exploring the transfer of those fatty acids to cord blood and any associations with neuropsychological effects in children?

Response to the Reviewer: Thanks for the comment, we added more clarifications in this paragraph and also two additional studies in order to justify the potential biological pathways linked to our findings. See changes in Discussion section, lines 429-431 and lines 436-438.

  1. Line 385, the sentence has grammatical errors and it is unclear.

“However, the complexity of human brain growth during the entire gestational period and its behavioural consequences in a long term are difficult to predict and there is gown for further scientific discussions of the potential biologic mechanisms...”

Response to the Reviewer: Thank you for this comment, our previous Reviewer also commented the same grammar problem. So, changes were added previously in Discussion section, lines 442-445.

Reviewer 3 Report

A few notes for possible improvement of an excellent paper:
Line 124) could be confusing. I would ask to Authors for a better definition of the outcomes "after short-term" and "long-term" intakes.

I would ask the authors for a comment in Discussion
Line 291) on the mercury content related to seafood, for the possible clinical implications, and
Line 289) on the discrepancy between dietary intakes at the 3rd trimester and values ​​in children in the subgroup (n. 948)

Author Response

Reviewer-3

A few notes for possible improvement of an excellent paper:

Line 124) could be confusing. I would ask to Authors for a better definition of the outcomes "after short-term" and "long-term" intakes.

Response to the Reviewer: Ok, thanks for the comment, we have added this clarification. Lines 130-131.

I would ask the authors for a comment in Discussion
Line 291) on the mercury content related to seafood, for the possible clinical implications,

Response to the Reviewer: Ok, as the previous Reviewer asked the same suggestion, we added this clinical implication in lines 487-495.

and
Line 289) on the discrepancy between dietary intakes at the 3rd trimester and values ​​in children in the subgroup (n. 948)

Response to the Reviewer: Ok, we have added a comment on this, see changes in lines 477-481.

Round 2

Reviewer 1 Report

                The authors have addressed a number of concerns and have taken a more cautious approach to their conclusions.  Including data from multiple sites increases the n, but makes uniformity of evaluations a challenge.   Epidemiological studies are by nature complex and open to bias at every stage. This study is based on a FFQ and such dietary measures are notoriously inaccurate.  Determining PUFA exposure from them adds another layer of possible inaccuracy.  

                The manuscript would benefit from careful review by a native English speaker for clarity of wording and grammar.  

Specifics:

Line 70                  I suggest shown be replaced by found.  It may be spurious? 

Line 112               Suggest you reverse supplementation and food sources or just go with preformed. 

Line 160               Suggest - the participants in this study

Line 161               Suggest - eligible women who present for…

Line 173               Suggest 92.4/% respectively of the…

Line 184               Suggest …was administered.  Would suggest

Line 204               This additional paragraph is less than reassuring.  The first sentence seems incomplete.  The next on validity correlations does not seem to have any relation to determining PUFA values.  The next doesn’t note what is being correlated?

Line 218               This sentence should be revised.  Usually covariates are selected for their impact on endpoints rather than being based on study questionnaires. 

Line 221               It appears the authors evaluated a number of other nutrients, aMED diet, Hg and maternal mental health suggesting they carried out substantially more analyses than the title suggests.

Line 290                Epidemiological studies are usually based on a biological hypothesis and not on what shows the “…most significant results”.        

Results line 336                 Having described the BSID, its association deserve a comment.

Discussion           It would benefit the reader to refer to studies using the first author and year

Line 389               Since the majority of n-3 are transferred in the 3rd trimester a comment on this discrepancy would be helpful

Lines 404-406    This perhaps fits better in the first paragraph describing the study’s findings.

Lines 413-414    The grammar starts with multiple and then changed to two.  Suggest rewording

Line 417               Needs a reference

Line 429               This sentence has 2 ideas and the second is unclear

 Line 449              Suggest rewording sentence

Line 461                Suggest grammar correction

Line 487               This paragraph seems out of place and does not clarify or help the study.  Offering advice on what was not studied is inappropriate.

Line 496               Suggest found instead of shows and improved for higher (some tests are better with lower scores)

Line 500               Please correct the grammar

Author Response

The authors have addressed a number of concerns and have taken a more cautious approach to their conclusions.  Including data from multiple sites increases the n, but makes uniformity of evaluations a challenge.   Epidemiological studies are by nature complex and open to bias at every stage. This study is based on a FFQ and such dietary measures are notoriously inaccurate.  Determining PUFA exposure from them adds another layer of possible inaccuracy.  

Response to the Reviewer: Thanks, we agree with the Reviewer and we clearly stated these limitations in the discussion section.

The manuscript would benefit from careful review by a native English speaker for clarity of wording and grammar.  

Response to the Reviewer: Thank you, the first author and second authors are Canadians and they have checked again for English grammar.

Specifics:

Line 70                  I suggest shown be replaced by found.  It may be spurious? 

Response to the Reviewer: Ok, we changed it in line 69.

Line 112               Suggest you reverse supplementation and food sources or just go with preformed. 

Response to the Reviewer: Ok, changed added in lines 108-111.

Line 160               Suggest - the participants in this study

Response to the Reviewer: Ok, changed added in line 154.

Line 161               Suggest - eligible women who present for…

Response to the Reviewer: Ok, changed added in line 155.

Line 173               Suggest 92.4/% respectively of the…

Response to the Reviewer: Ok, changed added in line 167.

Line 184               Suggest …was administered.  Would suggest

Response to the Reviewer: Ok, changed added in line 177.

Line 204               This additional paragraph is less than reassuring.  The first sentence seems incomplete.  The next on validity correlations does not seem to have any relation to determining PUFA values.  The next doesn’t note what is being correlated?

Response to the Reviewer: We apologize for the mistake that was due mainly when cutting and pasting the proposed text for the new version of the manuscript. We initially provided the information focusing on omega 3 (O3) fatty acid and now, we provide more detailed information on reproducibility/validity of the FFQ for the intake of PUFA and Omega 3 (O3, n-3 fatty acids). We propose to include the following text for the manuscript:

New text for the manuscript, lines 197-205:

“The overall reproducibility and validity of the FFQ was acceptable for the intake of most nutrients and food groups. The reproducibility of the FFQ for the intake of PUFA (r=0.50) and the n-3 PUFA (r=0.53) was acceptably good (18). The biochemical validity of the FFQ was also assessed in pregnant women of the INMA Sabadell cohort study (20); statistically significant correlations (p<0.01) were observed between plasma concentrations and dietary intakes for n-3 PUFA (r=0.19), EPA (r=0.33) and DHA (r=0.28). Thus, the overall validity of the FFQ to assess n-3 PUFA intake may be considered as acceptable. Lower correlations are usually found in the literature when comparing nutrient intakes with plasma concentrations (21).”  

Line 218               This sentence should be revised.  Usually covariates are selected for their impact on endpoints rather than being based on study questionnaires. 

Response to the Reviewer: We agree and we added the appropriate changes to make it clear. See changes in lines 219-222.

Line 221               It appears the authors evaluated a number of other nutrients, aMED diet, Hg and maternal mental health suggesting they carried out substantially more analyses than the title suggests.

Response to the Reviewer: Well, thank you, we agree that we carried out a bunch of secondary analyses in order to assess the independency to the main exposure association with the outcomes. We usually do those kinds of secondary analyses in epidemiological studies. However, these secondary analyses are good to be reported in the MS, but we consider important that the title should be preserved and focused with the main aim to the paper.

Line 290                Epidemiological studies are usually based on a biological hypothesis and not on what shows the “…most significant results”.     

Response to the Reviewer: Well, we agree on that, thanks. Actually, the two periods are biologically important. The problem is that due to length and extension of the MS, we rather prefer to show the none-significant results in supplementary material. In order to not confuse the reader, we have changed a bit the sentence. See changes in lines 290-291.

Results line 336                 Having described the BSID, its association deserve a comment.

Response to the Reviewer: Ok, we have commented it in the text. See changes in lines 338-340.

Discussion           It would benefit the reader to refer to studies using the first author and year

Response to the Reviewer: Ok, we have added this information with the main studies we referred.

Line 389               Since the majority of n-3 are transferred in the 3rd trimester a comment on this discrepancy would be helpful

Response to the Reviewer: OK, se have added this statement in lines 379-381

Lines 404-406    This perhaps fits better in the first paragraph describing the study’s findings.

Response to the Reviewer: Thank you for this comment, but we believe that if we include these lines in the previous paragraph, in which we described the present study findings, could create confusion to the reader and may be thinking that this other study results, described in these lines, would be part of our present study findings. So, we prefer to keep it as it is now.

Lines 413-414    The grammar starts with multiple and then changed to two.  Suggest rewording

Response to the Reviewer: Thank you, we changed it, see line 404.

Line 417               Needs a reference

Response to the Reviewer: Ok, done in line 408.

Line 429               This sentence has 2 ideas and the second is unclear

Response to the Reviewer: Ok, we agree, we have re-written the sentence. See changes in line 421-424.

 Line 449              Suggest rewording sentence

Response to the Reviewer: Ok, we agree, we have re-written the sentence. Se changes in line 441-442

Line 461                Suggest grammar correction

Response to the Reviewer: Ok, correction done in line 449-450.

Line 487               This paragraph seems out of place and does not clarify or help the study.  Offering advice on what was not studied is inappropriate.

Response to the Reviewer: Ok, we have deleted the full paragraph, lines 476-484.

Line 496               Suggest found instead of shows and improved for higher (some tests are better with lower scores)

Response to the Reviewer: Ok, done. Line 485.

Line 500               Please correct the grammar

Response to the Reviewer: Ok, done. Lines 489-491.

Reviewer 2 Report

My comments were adequately addressed; however, the manuscript still has some typos and grammatical errors, for example figure 2. I recommend publication after correcting those errors.

Author Response

My comments were adequately addressed; however, the manuscript still has some typos and grammatical errors, for example figure 2. I recommend publication after correcting those errors.

Response to the Reviewer: Ok, we have added English grammar corrections in the Figure 2 and the whole MS text. The first and second authors are Canadians and checked again all the potential grammar mistakes.
